# GNNGUARD: Defending Graph Neural Networks against Adversarial Attacks

**Xiang Zhang**
Harvard University
xiang_zhang@hms.harvard.edu

**Marinka Zitnik**
Harvard University
marinka@hms.harvard.edu

## Abstract

Deep learning methods for graphs achieve remarkable performance across a variety of domains. However, recent findings indicate that small, unnoticeable perturbations of graph structure can catastrophically reduce performance of even the strongest and most popular Graph Neural Networks (GNNs). Here, we develop GNNGUARD, a general algorithm to defend against a variety of training-time attacks that perturb the discrete graph structure. GNNGUARD can be straightforwardly incorporated into any GNN. Its core principle is to detect and quantify the relationship between the graph structure and node features, if one exists, and then exploit that relationship to mitigate negative effects of the attack. GNN-GUARD learns how to best assign higher weights to edges connecting similar nodes while pruning edges between unrelated nodes. The revised edges allow for robust propagation of neural messages in the underlying GNN. GNNGUARD introduces two novel components, the neighbor importance estimation, and the layer-wise graph memory, and we show empirically that both components are necessary for a successful defense. Across five GNNs, three defense methods, and four datasets, including a challenging human disease graph, experiments show that GNNGUARD outperforms existing defense approaches by 15.3% on average. Remarkably, GN-NGUARD can effectively restore state-of-the-art performance of GNNs in the face of various adversarial attacks, including targeted and non-targeted attacks, and can defend against attacks on heterophily graphs.

## 1 Introduction

Deep learning on graphs and Graph Neural Networks (GNNs), in particular, have achieved remarkable success in a variety of application areas [1, 2, 3, 4, 5]. The key to the success of GNNs is the neural message passing scheme [6] in which neural messages are propagated along edges of the graph and typically optimized for performance on a downstream task. In doing so, the GNN is trained to aggregate information from neighbors for every node in each layer, which allows the model to eventually generate representations that capture useful node feature as well as topological structure information [7]. While the aggregation of neighbor nodes' information is a powerful principle of representation learning, the way that GNNs exchange that information between nodes makes them vulnerable to adversarial attacks [8].

Adversarial attacks on graphs, which carefully rewire the graph topology by selecting a small number of edges or inject carefully designed perturbations to node features, can contaminate local node neighborhoods, degrade learned representations, confuse the GNN to misclassify nodes in the graph, and can catastrophically reduce the performance of even the strongest and most popular GNNs [9, 10]. The lack of GNN robustness is a critical issue in many application areas, including those where adversarial perturbations can undermine public trust [11], interfere with human decision making [12], and affect human health and livelihoods [13]. For this reason, it is vital to develop GNNs that are

robust against adversarial attacks. While the vulnerability of machine learning methods to adversarial attacks has raised many concerns and has led to theoretical insights into robustness [14] and the development of effective defense techniques [9, 12, 15], adversarial attacks and defense on graphs remain poorly understood.

**Present work.** Here, we introduce GNNGUARD[1], an approach that can defend any GNN model against a variety of training-time attacks that perturb graph structure (Figure 1). GNNGUARD takes as input an existing GNN model. It mitigates adverse effects by modifying the GNN's neural message passing operators. In particular, it revises the message passing architecture such that the revised model is robust to adversarial perturbations while at the same time the model keeps it representation learning capacity. To this end, GNNGUARD develops two key components that estimate neighbor importance for every node and coarsen the graph through an efficient memory layer. The former component dynamically adjusts the relevance of nodes' local network neighborhoods, prunes likely fake edges, and assigns less weight to suspicious edges based on network theory of homophily [16]. The latter components stabilizes the evolution of graph structure by preserving, in part the memory from a previous layer in the GNN.

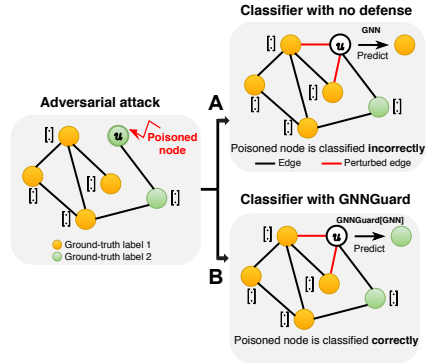

Figure 1: **A**. Small, adversarial perturbations of the graph structure and node features lead GNN to misclassify target $u$. **B**. The GNN, when integrated with GNNGUARD, correctly predicts $u$'s label.

We compare GNNGUARD to three state-of-the-art GNN defenders across four datasets and under a variety of attacks, including direct targeted, influence targeted, and non-targeted attacks. Experiments show that GNNGUARD improves state-of-the-art methods by up to 15.3% in defense performance. Importantly, unlike existing GNN defenders [17, 18, 19, 20], GNNGUARD is a general approach and can be effortlessly combined with any GNN architecture. To that end, we integrate GNNGUARD into five GNN models. Remarkably, results show that GNNGUARD can effectively restore state-of-the-art performance of even the strongest and most popular GNNs [3, 21, 7, 22, 23], thereby demonstrating broad applicability and relevance of GNNGUARD for graph machine-learning. Finally, GNNGUARD is the first technique that shows a successful defense on heterophily graphs [24]. In contrast, previous defenders, *e.g.*, [17, 18, 19, 20], focused on homophily graphs [16]. Results show that GNNGUARD can be easily generalized to graphs with abundant structural equivalences where connected nodes can have different node features yet similar structural roles within their local topology [25].

## 2  Related Work

**Adversarial attacks in continuous and discrete space.** Adversarial attacks on machine learning have received increasing attention in recent years [14, 26, 27]. The attackers add small perturbations on the samples to completely alter the output of the machine learning model. The deliberately manipulated perturbations are often designed to be unnoticeable. Modern studies have shown that machine leaning models, especially deep neural networks, are highly fragile to adversarial attacks [13, 28, 29]. The majority of existing works focus on grid data or independent samples [30] whilst a few work investigate adversarial attack on graphs.

**Adversarial attacks on graphs.** Based on the goal of the attacker, adversarial attacks on graphs [31, 32] can be divided into poisoning attacks (*e.g.*, Nettack [8]) that perturb the graph in training-time and evasion attacks (*e.g.*, RL-S2V [32]) that perturb the graph in testing-time. GNNGUARD is designed to improve robustness of GNNs against poisoning attacks. There are two types of poisoning attacks: a targeted attack and a non-targeted attack [33]. The former deceives the model to misclassify a specific node (*i.e.*, target node) [8] while the latter degrades the overall performance of the trained model [30]. The targeted attack can be categorized into direct targeted attack where the attacker perturbs edges touching the target node and the influence targeted attack where the attacker only manipulates edges of the target node's neighbors. Nettack [8] generates perturbations by modifying graph structure (*i.e.*, structure attack) and node attributes (*i.e.*, feature attack) such that perturbations maximally destroy

downstream GNN's predictions. Bojcheshki *et al.* [34] derive adversarial perturbations that poison the graph structure. Similarly, Zügner *et al.* [30] propose a non-targeted poisoning attacker by using meta-gradient to solve bi-level problem. In contrast, our GNNGUARD is a defense approach that inspects the graph and recovers adversarial perturbations.

**Defense on graphs.** While deep learning on graphs has shown exciting results in a variety of applications [6, 23, 35, 36, 37], little attention has been paid to the robustness of such models, in contrast to an abundance of research for image (*e.g.*, [38]) and text (*e.g.*, [39]) adversarial defense. We briefly overview the state-of-the-art defense methods on graphs. GNN-Jaccard [17] is a defense approach that pre-processes the adjacency matrix of the graph to identify the manipulated edges. While GNN-Jaccard can defend targeted adversarial attacks on known and already existing GNNs, there has also been work on novel, robust GNN models. For example, RobustGCN [19] is a novel GNN that adopts Gaussian distributions as the hidden representations of nodes in each convolutional layer to absorb the effect of an attack. Similarly, GNN-SVD [18] uses a low-rank approximation of adjacency matrix that drops noisy information through an SVD decomposition. Tang *et al.* [20] improve the robustness of GNNs against poisoning attack through transfer learning but has a limitation that requires several unperturbed graphs from the similar domain during training. However, all these approaches have drawbacks (see Section 4.3) that prevent them from realizing their potential for defense to the fullest extent. For instance, none of them consider how to defend heterophily graphs against adversarial attacks. GNNGUARD eliminates these drawbacks, successfully defending targeted and non-targeted poisoning attacks on any GNN without decreasing its accuracy.

## 3  Background and Problem Formulation

Let $\mathcal{G} = (\mathcal{V}, \mathcal{E}, \mathbf{X})$ denote a graph where $\mathcal{V}$ is the set of nodes, $\mathcal{E}$ is the set of edges and $\mathbf{X} = \{\mathbf{x}_1, ..., \mathbf{x}_n\}, \mathbf{x}_u \in \mathbb{R}^M$ is the $M$-dimensional node feature for node $u \in \mathcal{V}$. Let $N = |\mathcal{V}|$ and $E = |\mathcal{E}|$ denote the number of nodes and edges, respectively. Let $\boldsymbol{A} \in \mathbb{R}^{N \times N}$ denote an adjacency matrix whose element $\boldsymbol{A}_{uv} \in \{0, 1\}$ indicates existence of edge $e_{uv}$ that connects node $u$ and $v$. We use $\mathcal{N}_u$ to denote immediate neighbors of node $u$, including the node itself ($u \in \mathcal{N}_u$). We use $\mathcal{N}_u^*$ to indicate $u$'s neighborhood, excluding the node itself ($u \notin \mathcal{N}_u^*$). Without loss of generality, we consider node classification task, wherein a GNN $f$ classifies nodes into $C$ labels. Let $\hat{y}_u = f_u(\mathcal{G})$ denote prediction for node $u$, and let $y_u \in \{1, \ldots, C\}$ denote the associated ground-truth label for node $u$. To degrade the performance of $f$, an adversarial attacker perturbs edges in $\mathcal{G}$, resulting in the perturbed version of $\mathcal{G}$, which we call $\mathcal{G}' = (\mathcal{V}, \mathcal{E}', \mathbf{X})$ ($\mathbf{A}'$ is adjacency matrix of $\mathcal{G}'$).

**Background on graph neural networks.** Graph neural networks learns compact, low-dimensional representations, *i.e.*, embeddings, for nodes such that representation capture nodes' local network neighborhoods as well as nodes' features [6, 3, 40]. The learned embeddings can be used for a variety of downstream tasks [3]. Let $\boldsymbol{h}_u^k \in \mathbb{R}^{D_k}$ denote the embedding of node $u$ in the $k$-th layer of GNN, $k = \{1, \ldots, K\}$. The $D_k$ stands for the dimension of $\boldsymbol{h}_u^k$. Note that $\boldsymbol{h}_u^0 = \boldsymbol{x}_u$. The computations in the $k$-th layer consist of a message-passing function MSG, an aggregation function AGG, and an update function UPD. This means that a GNN $f$ can be specified as $f = (\text{MSG}, \text{AGG}, \text{UPD})$ [6, 36]. Given a node $u$ and its neighbor $v \in \mathcal{N}_u$, the messaging-passing function MSG specifies what neural message $\boldsymbol{m}_{uv}^k$ needs to be propagated from $v$ to $u$. The message is calculated by $\boldsymbol{m}_{uv}^k = \text{MSG}(\boldsymbol{h}_u^k, \boldsymbol{h}_v^k, \boldsymbol{A}_{uv})$, where MSG receives node embeddings of $u$ and $v$ along with their connectivity information $e_{uv}$. This is followed by the aggregation function AGG that aggregates all messages received by $u$. The aggregated message $\hat{\boldsymbol{m}}_u^k$ is computed by $\hat{\boldsymbol{m}}_u^k = \text{AGG}(\{\boldsymbol{m}_{uv}^k; v \in \mathcal{N}_u^*\})$. Lastly, the update function UPD combines $u$'s embedding $\boldsymbol{h}_u^k$ and the aggregated message $\hat{\boldsymbol{m}}_u^k$ to generate the embedding for next layer as $\boldsymbol{h}_u^{k+1} = \text{UPD}(\boldsymbol{h}_u^k, \hat{\boldsymbol{m}}_u^k)$. The final node representation for $u$ is $\boldsymbol{h}_u^K$, *i.e.*, the output of the $K$-th layer.

**Background on poisoning attacks.** Attackers try to fool a GNN by corrupting the graph topology during training [41]. The attacker carefully selects a small number of edges and manipulates them through perturbation and rewiring. In doing so, the attacker aims to fool the GNN into making incorrect predictions [20]. The attacker finds optimal perturbation $\boldsymbol{A}'$ through optimization [30, 8]:

$$\underset{\boldsymbol{A}' \in \mathcal{P}_\Delta^\mathcal{G}}{\text{argmin}} \, \mathcal{L}_{\text{attack}}(f(\boldsymbol{A}', \boldsymbol{X}; \Theta^*), \boldsymbol{y}) \quad \text{s.t.} \quad \Theta^* = \underset{\Theta}{\text{argmin}} \, \mathcal{L}_{\text{predict}}(f(\boldsymbol{A}', \boldsymbol{X}; \Theta), \boldsymbol{y}) \quad (1)$$

where $\boldsymbol{y}$ denotes ground-truth labels, $\mathcal{L}_{\text{attack}}$ denotes the attacker's loss function, and $\mathcal{L}_{\text{predict}}$ denotes GNN's loss. The $\Theta^*$ refers to optimal parameters and $f(\boldsymbol{A}', \boldsymbol{X}; \Theta^*)$ is prediction of $f$ with parameters

$\Theta^*$ on the perturbed graph $\boldsymbol{A}'$ and node features $\boldsymbol{X}$. To ensure that attacker perturbs only a small number of edges, a budget $\Delta$ is defined to constrain the number of perturbed edges: $||\boldsymbol{A}' - \boldsymbol{A}||_0 \leqslant \Delta$ and $\mathcal{P}_\Delta^\mathcal{G}$ are perturbations that fit into budget $\Delta$. Let $\mathcal{T}$ be target nodes that are intended to be mis-classified, and let $\mathcal{A}$ be attacker nodes that are allowed to be perturbed. We consider three types of attacks. (1) Direct targeted attacks. The attacker aims to destroy prediction for target node $u$ by manipulating the incident edges of $u$ [8, 17]. Here, $\mathcal{T} = \mathcal{A} = \{u\}$. (2) Influence targeted attacks. The attacker aims to destroy prediction for target node $u$ by perturbing the edges of $u$'s neighbors. Here, $\mathcal{T} = \{u\}$ and $\mathcal{A} = \mathcal{N}_u^*$. (3) Non-targeted attacks. The attacker aims to degrade overall GNN classification performance [30, 42]. Here, $\mathcal{T} = \mathcal{A} = \mathcal{V}_{\text{test}}$ where $\mathcal{V}_{\text{test}}$ denotes the test set.

## 3.1 GNNGUARD: Problem Formulation

GNNGUARD is a defense mechanism that is easy to integrate into any GNN $f$, resulting in a new GNN $f'$ that is robust to poisoning attacks. This means that $f'$ can make correct predictions even when trained on poisoned graph $\mathcal{G}'$. Given a GNN $f = (\text{MSG}, \text{AGG}, \text{UPD})$, GNNGUARD will return a new GNN $f' = (\text{MSG}', \text{AGG}', \text{UPD}')$, where $\text{MSG}'$ is the message-passing function, $\text{AGG}'$ is the aggregation function, and $\text{UPD}'$ is the update function. The $f'$ solves the following defense problem.

**Problem (Defense Against Poisoning Attacks on Graphs).** *In a poisoning attack, the attacker injects adversarial edges in $\mathcal{G}$, meaning that the attack changes training data, which can decrease the performance of GNN considerably. Let $\mathcal{G}'$ denote the perturbed version of $\mathcal{G}$ that is poisoned by the attack. We seek GNN $f'$ such that for any node $u \in \mathcal{G}'$:*

$$min \ f_u'(\mathcal{G}') - f_u(\mathcal{G}), \tag{2}$$

*where $f_u'(\mathcal{G}') = \hat{y}_u'$ is the prediction when GNN $f'$ is trained on $\mathcal{G}'$. Here, $f_u(\mathcal{G}) = \hat{y}_u$ denotes a hypothetical prediction that the GNN would made if it had access to clean graph $\mathcal{G}$.*

It is worth noting that, in this paper, we learn a defense mechanism for semi-supervised node classification. GNNGUARD is a general framework for defending any GNN on various graph mining tasks such as link prediction. Since there exists a variety of GNNs that achieve competitive performance on $\mathcal{G}$, an intuitive idea is to force $f_u'(\mathcal{G}')$ to approximate $f_u(\mathcal{G})$ and, in doing so, ensure that $f'$ will make correct predictions on $\mathcal{G}'$. For this reason, we design $f'$ to learn neural messages on $\mathcal{G}'$ that, in turn, are similar to the messages that a hypothetical $f$ would learn on $\mathcal{G}$. However, since it is impossible to access clean graph $\mathcal{G}$, Eq. (2) can not be directly optimized. The key to restore the structure of $\mathcal{G}$ is to design a message-passing scheme that can detect fake edges, block them and then attend to true, unperturbed edges. To this end, the impact of perturbed edges in $\mathcal{G}'$ can be mitigated by manipulating the flow of neural messages and thus, the structure of $\mathcal{G}$ can be restored.

## 4 GNNGUARD

Next, we describe GNNGUARD, our GNN defender against poisoning attacks. Recent studies [31, 17] found that most damaging attacks add fake edges between nodes that have different features and labels. Because of that, the core defense principle of GNNGUARD is to detect such fake edges and alleviate their negative impact on prediction by remove them or assigning them lower weights in neural message passing. GNNGUARD has two key components: (1) neighbor importance estimation, and (2) layer-wise graph memory, the first component being an essential part of a robust GNN architecture while the latter is designed to smooth the defense.

### 4.1 Neighbor Importance Estimation

GNNGUARD estimates an importance weight for every edge $e_{uv}$ to quantify how relevant node $u$ is to another node $v$ in the sense that it allows for successful routing of GNN's messages. In contrast to attention mechanisms (*e.g.*, GAT [21, 43]), GNNGUARD determines importance weights based on the hypothesis that similar nodes (*i.e.*, nodes with similar features or similar structural roles) are more likely to interact than dissimilar nodes [16]. To this end, we quantify similarity $s_{uv}^k$ between $u$ and its neighbor $v$ in the $k$-th layer of GNN as follows:

$$s_{uv}^k = d(\boldsymbol{h}_u^k, \boldsymbol{h}_v^k), \qquad d(\boldsymbol{h}_u^k, \boldsymbol{h}_v^k) = (\boldsymbol{h}_u^k \odot \boldsymbol{h}_v^k)/(||\boldsymbol{h}_u^k||_2 ||\boldsymbol{h}_v^k||_2), \tag{3}$$

where $d$ is a similarity function and $\odot$ denotes dot product. In this work, we use cosine similarity to calculate $d$ [44]. In homophily graphs, $s_{uv}^k$ measures the similarity between node features; in

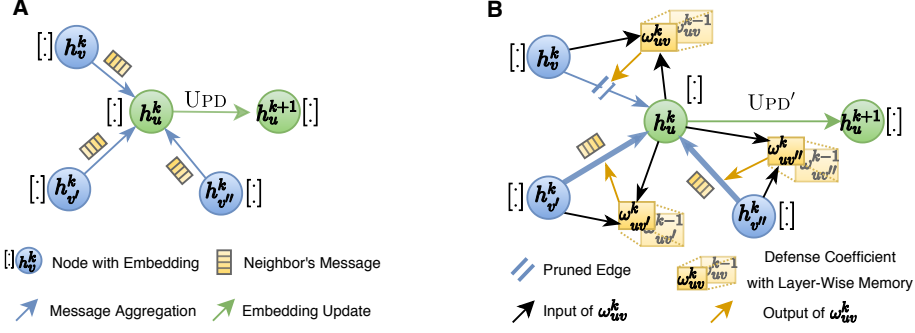

**Figure 2: A.** Illustration of neural message passing in $u$'s local network neighborhood in the $k$-th layer of GNN $f$. **B.** The message flow in $f'$, which is the GNN $f$ endowed by GNNGUARD defense. We first calculate defense coefficients $\boldsymbol{\omega}_{uv}^k$ based on node representations $\boldsymbol{h}_u^k$ and $\boldsymbol{h}_v^k$. The defense coefficients are then used to control the message stream such as blocking the message from $v$ but strengthening messages from $v'$ and $v''$. Thick blue arrow indicates the higher weights during message aggregation. To stabilize the evolution of graph structure, current defense coefficients (*e.g.*, $\boldsymbol{\omega}_{uv}^k$) keep a partial memory of the previous layer (*e.g.*, $\boldsymbol{\omega}_{uv}^{k-1}$).

heterophily graphs, it measures the similarity of nodes' structural roles. Larger similarity $s_{uv}^k$ indicates that edge $e_{uv}$ is strongly supported by node features (or local topology) of the edge's endpoints. We normalize $s_{uv}^k$ at the node-level within $u$'s neighborhood $\mathcal{N}_u$. The problem here is to specify what is the similarity of the node to itself. We normalize node similarities as:

$$\alpha_{uv}^k = \begin{cases} s_{uv}^k / \sum_{v \in \mathcal{N}_u^*} s_{uv}^k \times \hat{N}_u^k / (\hat{N}_u^k + 1) & \text{if} \quad u \neq v \\ 1/(\hat{N}_u^k + 1) & \text{if} \quad u = v, \end{cases} \tag{4}$$

where $\hat{N}_u^k = \sum_{v \in \mathcal{N}_u^*} ||s_{uv}^k||_0$. We refer to $\alpha_{uv}^k$ as an importance weight representing the contribution of node $v$ towards node $u$ in the GNN's passing of neural messages in poisoned graph $\mathcal{G}'$. In doing so, GNNGUARD assigns small importance weights to suspicious neighbors, which reduce the interference of suspicious nodes in GNN's operation. Further, to alleviate the impact of fake edges, we prune edges that are likely forged. Building on network homophily and findings [17] that fake edges tend to connect dissimilar nodes, we prune edges using importance weights. For that, we define a characteristic vector $\boldsymbol{c}_{uv}^k = [\alpha_{uv}^k, \alpha_{vu}^k]$ describing edge $e_{uv}$. Although $s_{uv}^k = s_{vu}^k$, it is key to note that $\alpha_{uv}^k \neq \alpha_{vu}^k$ because of self-normalization in Eq. (4). GNNGUARD calculates edge pruning probability for $e_{uv}$ through a non-linear transformation as $\sigma(\boldsymbol{c}_{uv}^k \boldsymbol{W})$. Then, it maps the pruning probability to a binary indicator $1_{P_0} : \sigma(\boldsymbol{c}_{uv}^k \boldsymbol{W})$ where $P_0$ is a user-defined threshold:

$$1_{P_0}(\sigma(\boldsymbol{c}_{uv}^k \boldsymbol{W})) = \begin{cases} 0 & \text{if} \quad \sigma(\boldsymbol{c}_{uv}^k \boldsymbol{W}) < P_0 \\ 1 & \text{otherwise.} \end{cases} \tag{5}$$

Finally, we prune edges by updating importance weight $\alpha_{uv}^k$ to $\hat{\alpha}_{uv}^k$ as follows:

$$\hat{\alpha}_{uv}^k = \alpha_{uv}^k 1_{P_0}(\sigma(\boldsymbol{c}_{uv}^k \boldsymbol{W})), \tag{6}$$

meaning that the perturbed edges connecting dissimilar nodes will likely be ignored by the GNN.

## 4.2 Layer-Wise Graph Memory

Neighbor importance estimation and edge pruning change the graph structure between adjacent GNN layers. This can destabilize GNN training, especially if a considerable number of edges gets pruned in a single layer (*e.g.*, due to the weight initialization). To allow for robust estimation of importance weights and smooth evolution of edge pruning, we use layer-wise graph memory. This unit, applied at each GNN layer, keeps partial memory of the pruned graph structure from the previous layer (Figure 2). We define layer-wise graph memory as follows:

$$\omega_{uv}^k = \beta \omega_{uv}^{k-1} + (1 - \beta)\hat{\alpha}_{uv}^k, \tag{7}$$

where $\omega_{uv}^k$ represents defense coefficient for edge $e_{uv}$ in the $k$-th layer and $\beta$ is a memory coefficient specifying memory, *i.e.*, the amount of information from the previous layer that should be kept in the current layer. Memory coefficient $\beta \in [0, 1]$ is a learnable parameter and is set to $\beta = 0$ in the first GNN layer, meaning that $\omega_{uv}^0 = \hat{\alpha}_{uv}^0$. Using defense coefficients, GNNGUARD controls information flow across all neural message passing layers. It strengthens messages from $u$'s neighbors with higher defense coefficients and weakens messages from $u$'s neighbors with lower defense coefficients.

## 4.3 Overview of GNNGUARD

GNNGUARD is shown in Algorithm 1. The method is easy to plug into an existing GNN to defend the GNN against poisoning attacks. Given a GNN $f = (\text{MSG}, \text{AGG}, \text{UPD})$, GNNGUARD formulates a revised version of it, called $f' = (\text{MSG}', \text{AGG}', \text{UPD}')$. In each layer, $f'$ takes current node representations and (possibly attacked) graph $\mathcal{G}'$. It estimates importance weights $\hat{\alpha}_{uv}$ and generates defense coefficients $\omega_{uv}$ by combining importance weights from the current layer and defense coefficients from the previous layer. In summary, aggregation function $\text{AGG}'$ in layer $k$ is: $\text{AGG}' = \text{AGG}(\{\omega_{uv}^k \odot \boldsymbol{m}_{uv}^k; v \in \mathcal{N}_u^*\})$. The update function $\text{UPD}'$ is: $\text{UPD}' = \text{UPD}(\omega_{uu}^k \odot \boldsymbol{h}_u^k, \text{AGG}(\{\omega_{uv}^k \odot \boldsymbol{m}_{uv}^k; v \in \mathcal{N}_u^*\}))$. The message function $\text{MSG}'$ remains unchanged $\text{MSG}' = \text{MSG}$ as neural messages are specified by the original GNN $f$. Taken together, the guarded $f'$ attends differently to different node neighborhoods and propagates neural information only along most relevant edges. Our derivations here are for undirected graphs with node features but can be extended to directed graphs and edge features (*e.g.*, include them into calculation of characteristic vectors).

---

**Algorithm 1:** GNNGUARD.

---

**Input**: GNN model of interest $f = (\text{MSG}, \text{AGG}, \text{UPD})$; Poisoned graph $\mathcal{G}' = (\mathcal{V}, \mathcal{E}', \boldsymbol{X})$, ($\boldsymbol{A}'$ is adjacency matrix of $\mathcal{E}'$); Trainable parameters $\Theta, \boldsymbol{W}$, and $\beta$
Initialize parameters $\Theta, \boldsymbol{W}$, and $\beta$; initialize node representations $\boldsymbol{h}_u^0 = \boldsymbol{x}_u \; \forall u \in \mathcal{V}$
**for** layer $k \leftarrow 1$ **to** $K$ **do**
  **for** $u \in \mathcal{V}$ **do**
    Calculate $\alpha_{uv}^k$ using Eq. (4) for all $v \in \mathcal{N}_u$    // Neighbor Importance Estimation
    $\boldsymbol{c}_{uv}^k = [\alpha_{uv}^k, \alpha_{vu}^k]$
    $\hat{\alpha}_{uv}^k = \alpha_{uv}^k \mathbb{1}_{P_0}(\sigma(\boldsymbol{c}_{uv}^k \boldsymbol{W}))$ using Eq. (6)
    $\omega_{uv}^k = \beta \omega_{uv}^{k-1} + (1-\beta)\hat{\alpha}_{uv}^k$ using Eq. (7)        // Layer-Wise Graph Memory
    $\boldsymbol{m}_{uv}^k = \text{MSG}'(\boldsymbol{h}_u^k, \boldsymbol{h}_v^k, \boldsymbol{A}_{uv}')$ using Section 4.3       // Neural Message Passing
    $\hat{\boldsymbol{m}}_u^k = \text{AGG}'(\{\omega_{uv}^k \odot \boldsymbol{m}_{uv}^k; v \in \mathcal{N}_u^*\})$ using Section 4.3
    $\boldsymbol{h}_u^{k+1} = \text{UPD}'(\omega_{uu}^k \odot \boldsymbol{h}_u^k, \hat{\boldsymbol{m}}_u^k)$ using Section 4.3
  **end**
**end**

---

**Any GNN model.** State-of-the-art GNNs use neural message passing comprising of MSG, AGG, and UPD functions. As we demonstrate in experiments, GNNGUARD can defend such GNN architectures against adversarial attacks. GNNGUARD works with many GNNs, including Graph Convolutional Network (GCN) [3], Graph Attention Network (GAT) [21], Graph Isomorphism Network (GIN) [7], Jumping Knowledge (JK-Net) [22], GraphSAINT [23], GraphSAGE [40], and SignedGCN [45].

**Computational complexity.** GNNGUARD is practically efficient because it exploits the sparse structure of real-world graphs. The time complexity of neighbor importance estimation is $\mathcal{O}(D_k E)$ in layer $k$, where $D_k$ is the embedding dimensionality and $E$ is the graph size, and the complexity of layer-wise graph memory is $\mathcal{O}(E)$. This means that time complexity of GNNGUARD grows linearly with the size of the graph as node embeddings are low-dimensional, $D_k \ll E$. Finally, the time complexity of a GNN endowed with GNNGUARD is on the same order as that of the GNN itself.

**Further related work on adversarial defense for graphs.** We briefly contrast GNNGUARD with existing GNN defenders. Compared to GNN-Jaccard [17], which examines fake edges as a GNN preprocessing step, GNNGUARD dynamically updates defense coefficients at every GNN layer for defense. In contrast to RobustGCN [19], which is limited to GCN, a particular GNN variant, and is challenging to use with other GNNs, GNNGUARD provides a generic mechanism that is easy to use with many GNN architectures. Further, in contrast to GNN-SVD [18], which uses only graph structure for defense, GNNGUARD takes advantage of information encoded in both node features and graph structure. Also, [18] is designed specifically for the Nettack attacker [8] and so is less versatile. Another technique [20] uses transfer learning to detect fake edges. While that is an interesting idea, it requires a large number of clean graphs from the same domain to successfully train the transfer model. On the contrary, GNNGUARD takes advantage of correlation between node features and graph structure and does not need any external data. Further, recent studies (*e.g.*, [46, 47]) focus on theoretical certificates for GNN robustness instead of defense mechanisms. That is an important but orthogonal direction to this paper, where the focus is on a practical adversarial defense framework.

**Table 1:** Defense performance (multi-class classification accuracy) against direct targeted attacks.

| Model | Dataset | No Attack | Attack | GNN-Jaccard | RobustGCN | GNN-SVD | GNNGUARD |
|-------|---------|-----------|--------|-------------|-----------|---------|----------|
| GCN | Cora | 0.826 | 0.250 | 0.525 | 0.215 | 0.475 | **0.705** |
| | Citeseer | 0.721 | 0.175 | 0.435 | 0.230 | 0.615 | **0.720** |
| | ogbn-arxiv | 0.667 | 0.235 | 0.305 | 0.245 | 0.370 | **0.425** |
| | DP | 0.682 | 0.215 | 0.340 | 0.315 | 0.395 | **0.430** |
| GAT | Cora | 0.827 | 0.245 | 0.295 | 0.215 | 0.365 | **0.625** |
| | Citeseer | 0.718 | 0.265 | 0.575 | 0.230 | 0.575 | **0.765** |
| | ogbn-arxiv | 0.669 | 0.210 | 0.355 | 0.245 | 0.445 | **0.520** |
| | DP | 0.714 | 0.205 | 0.320 | 0.315 | 0.335 | **0.445** |
| GIN | Cora | 0.831 | 0.270 | 0.375 | 0.215 | 0.375 | **0.645** |
| | Citeseer | 0.725 | 0.285 | 0.570 | 0.230 | 0.570 | **0.755** |
| | ogbn-arxiv | 0.661 | 0.315 | 0.425 | 0.245 | 0.475 | **0.640** |
| | DP | 0.719 | 0.245 | 0.410 | 0.315 | 0.405 | **0.460** |
| JK-Net | Cora | 0.834 | 0.305 | 0.445 | 0.215 | 0.425 | **0.690** |
| | Citeseer | 0.724 | 0.275 | 0.615 | 0.230 | 0.610 | **0.775** |
| | ogbn-arxiv | 0.678 | 0.335 | 0.375 | 0.245 | 0.325 | **0.635** |
| | DP | 0.726 | 0.220 | 0.335 | 0.315 | 0.360 | **0.450** |
| Graph SAINT | Cora | 0.821 | 0.225 | 0.535 | 0.235 | 0.460 | **0.695** |
| | Citeseer | 0.716 | 0.195 | 0.470 | 0.350 | 0.395 | **0.770** |
| | ogbn-arxiv | 0.683 | 0.245 | 0.365 | 0.245 | 0.315 | **0.375** |
| | DP | 0.739 | 0.205 | 0.315 | 0.295 | 0.330 | **0.485** |

# 5 Experiments

We start by describing the experimental setup. We then present how GNNGUARD compares to existing GNN defenders (Section 5.1), provide an ablation study and a case study on citation network (Section 5.2), and show how GNNGUARD can be used with heterophily graphs (Section 5.3).

**Datasets.** We test GNNGUARD on four graphs. We use two citation networks with undirected edges and binary features: Cora [48] and Citeseer [49]. We also consider a directed graph with numeric node features, ogbn-arxiv [50], representing a citation network of CS papers published between 1971 and 2014. We use a Disease Pathway (DP) [51] graph with continuous features describing a system of interacting proteins whose malfunction collectively leads to diseases. The task is to predict for every protein node what diseases the protein might cause. Details are in Appendix D.

**Setup.** (1) Generating adversarial attacks. We compare our model to baselines under three kinds of adversarial attacks: direct targeted attack (Nettack-Di [8]), influence targeted attack (Nettack-In [8]), and non-targeted attack (Mettack [30]). In Mettack, we set the perturbation rate as $20\%$ (*i.e.*, $\Delta = 0.2E$) with 'Meta-Self' training strategy. In Nettack-Di, $\Delta = \hat{N}_u^0$. In Nettack-In, we perturb 5 neighbors of the target node and set $\Delta = \hat{N}_v^0$ for all neighbors. In the targeted attack, we select 40 correctly classified target nodes (following [8]): 10 nodes with the largest classification margin, 20 random nodes, and 10 nodes with the smallest margin. We run the whole attack and defense procedure for each target node and report average classification accuracy. (2) GNNs. We integrate GNNGUARD with five GNNs (GCN [3], GAT [21], GIN [7], JK-Net [22], and GraphSAINT [23]) and present the defense performance against adversarial attacks. (3) Baseline defense algorithms. We compare GNNGUARD to three state-of-the-art graph defenders: GNN-Jaccard [17], RobustGCN [19], and GNN-SVD [18]. Hyperparameters and model architectures are in Appendix E.

## 5.1 Results: Defense Against Targeted and Non-Targeted Attacks

(1) Results for direct targeted attacks. We observe in Table 1 that Nettack-Di is a strong attacker and dramatically cuts down the performance of all GNNs (cf. "Attack" vs. "No Attack" columns). However, the proposed GNNGUARD outperforms state-of-art defense methods by 15.3% in the accuracy on average. Further, it successfully restores the performance of GNNs to the level comparable to when there is no attack. We also observe that RobustGCN fails to defend against Nettack-Di, possibly because the Gaussian layer in RobustGCN cannot absorb big effects when all fake edges are in the vicinity of a target node. In contrast, GNN-SVD works well here because it is sensitive to high-rank noise caused by the perturbation of many edges that are incident to a single node. (2) Results for influence targeted attacks. As shown in Table 2, GNNGUARD achieves the best classification accuracy comparing to other baseline defense algorithms. Taking a closer look at the results, we we can find that Nettack-In is relatively less threaten than Nettack-Di indicating part of the perturbed information was scattered during neural message passing. (3) Results for non-targeted attacks. Table 2 shows that Mettack has a considerable negative impact on GNN performance, decreasing the

**Table 2:** Defense performance (multi-class classification accuracy) against influence targeted (top) and non-targeted (bottom) attacks. Tables with full results for other GNN models are in Appendix A (influence targeted attacks) and Appendix B (non-targeted attacks).

| Model | Dataset | No Attack | Attack | GNN-Jaccard | RobustGCN | GNN-SVD | GNNGUARD |
|---|---|---|---|---|---|---|---|
| GIN | Cora | 0.831 | 0.525 | 0.635 | 0.605 | 0.615 | **0.775** |
| | Citeseer | 0.725 | 0.480 | 0.675 | 0.575 | 0.630 | **0.845** |
| | ogbn-arxiv | 0.661 | 0.570 | 0.605 | 0.620 | 0.525 | **0.710** |
| | DP | 0.719 | 0.505 | 0.585 | 0.565 | 0.605 | **0.695** |
| GIN | Cora | 0.831 | 0.588 | 0.702 | 0.571 | 0.692 | **0.722** |
| | Citeseer | 0.725 | 0.565 | 0.638 | 0.583 | 0.615 | **0.711** |
| | ogbn-arxiv | 0.661 | 0.424 | 0.459 | 0.436 | 0.459 | **0.486** |
| | DP | 0.719 | 0.537 | 0.559 | 0.528 | 0.513 | **0.571** |

**Table 3:** Ablation study on ogbn-arxiv dataset. 'Memory' denotes layer-wise graph memory (Section 4.2) while 'pruning' denotes edge pruning operation (Section 4.1).

| Model | No Defense | GNNGUARD w/o pruning | GNNGUARD w/o memory | Full GNNGUARD |
|---|---|---|---|---|
| GCN | 0.235 | 0.350 | 0.405 | **0.425** |
| GAT | 0.210 | 0.315 | 0.475 | **0.520** |
| GIN | 0.315 | 0.540 | 0.610 | **0.640** |
| JK-Net | 0.335 | 0.565 | 0.625 | **0.635** |
| GraphSAINT | 0.245 | 0.305 | 0.360 | **0.375** |

accuracy of even the strongest GNN by 18.7% on average. Moreover, we see that GNNGUARD achieves a competitive performance and outperforms baselines in 19 out of 20 settings. In summary, experiments show the GNNGUARD consistently outperforms all baseline defense techniques. Further, GNNGUARD can defend a variety of GNNs against different types of attacks, indicating that GNNGUARD is a powerful GNN defender against adversarial poisoning.

## 5.2 Results: Ablation Study and Inspection of Defense Mechanism

(1) Ablation study. We conduct an ablation study to evaluate the necessity of every component of GNNGUARD. For that, we took the largest dataset (ogbn-arxiv) and the most threatening attack (Nettack-Di) as an example. Results are in Table 3. We observe that full GNNGUARD behaves better and has smaller standard deviation than limited GNNGUARD w/o layer-wise graph memory, suggesting that graph memory can contribute to defense performance and stabilize model training. (2) Node classification on clean datasets. In principle, we don't know if the input graph has been attacked or not. Because of that, it is important that a successful GNN defender can deal with poisoned graphs and also does not harm GNN performance on clean datasets. Appendix C shows classification accuracy of GNNs on clean graphs. Across all datasets, we see that, when graphs are not attacked, GNNs with turned-on GNNGUARD achieve performance comparable to that of GNNs alone, indicating that GNNGUARD will not weaken learning ability of GNNs when there is no attack. (3) Defense under different attack intensity. We investigate defense performance as a function of attack strength. Table 4 shows attack and defense results on Cora under Mettack with increasing attack rates. It is expected that GCN accuracy decays as attacks intensify. Nevertheless, GNNGUARD effectively defends GCN and can do so especially under strong attack. GCN with GNNGUARD outperforms GCN with no defense by 19.8% when 25% of the edges are attacked.

**A case study of attack and defense.** We report an example of attack and defense illustrating how GNNGUARD works. Let's examine the paper "TreeP: A Tree Based P2P Network Architecture" by Hudzia *et al.* [52] that received four citations. The topic/label of this paper (*i.e.*, node $u$ in ogbn-arxiv graph $\mathcal{G}$) and its cited works (*i.e.*, neighbors) is *Internet Technology (IT)*. GIN [7] trained on clean ogbn-arxiv graph makes a correct prediction for the paper with high confidence, $f_u(\mathcal{G}) = 0.536$. Then, we poison the paper using Nettack-Di attacker, which adds four fake citations between the paper and some very dissimilar papers from the *Artificial Intelligence (AI)* field. We re-trained GIN on perturbed graph $\mathcal{G}'$ and found the resulting classifier misclassifies the paper [52] into topic *AI* with confidence of $f_u(\mathcal{G}') = 0.201$, which is high on this prediction task with 40 distinct topics/labels. This fragility is especially worrisome as the attacker has only injected four fake citations and was already able to easily fool a state-of-the-art GNN. We then re-trained GIN with GNNGUARD defense on the same perturbed graph and, remarkably, the paper [52] was correctly classified to *IT* with high confidence ($f'_u(\mathcal{G}') = 0.489$) even after the attack. This example illustrates how easily an adversary can fool a GNN on citation networks.

**Table 4:** Attack and defense accuracy on Cora dataset.

| Attack Rate (% edges) | No Defense | GNNGUARD |
|---|---|---|
| 5% | 0.771 | **0.776** |
| 10% | 0.716 | **0.749** |
| 15% | 0.651 | **0.739** |
| 20% | 0.578 | **0.714** |
| 25% | 0.531 | **0.729** |

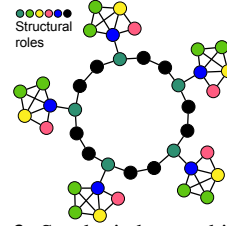

**Figure 3:** Synthetic heterophily graph. Node colors indicate structural roles.

**Table 5:** Performances of heterophily graphs. N/A means the method does not apply to the heterophily setting.

| Model | No Attack | Attack | GNN-Jaccard | RobustGCN | GNN-SVD | GNNGUARD |
|---|---|---|---|---|---|---|
| GCN | 0.834 | 0.385 | N/A | 0.525 | 0.595 | **0.715** |
| GAT | 0.851 | 0.325 | N/A | 0.575 | 0.635 | **0.770** |
| GIN | 0.891 | 0.450 | N/A | 0.575 | 0.650 | **0.775** |
| JK-Net | 0.889 | 0.425 | N/A | 0.575 | 0.640 | **0.735** |
| GraphSAINT | 0.876 | 0.415 | N/A | 0.575 | 0.625 | **0.755** |

## 5.3 Results: Defense of Heterophily Graphs

**GNNGUARD with heterophily.** Next, we evaluate GNNGUARD on graphs with structural roles, a prominent type of heterophily. To measure the local topology of nodes, we use graphlet degree vectors [53] which reflect nodes' structural properties, e.g., triangles, betweenness, stars, etc. To do so, we revise Eq. (3) by replacing embeddings for nodes $u$ and $v$ (*i.e.*, $\boldsymbol{h}_u^k$ and $\boldsymbol{h}_v^k$) with their graphlet degree vectors (*i.e.*, $\bar{\boldsymbol{h}}_u^k$ and $\bar{\boldsymbol{h}}_v^k$), yielding the learned similarity $s_{uv}^k$ that quantifies structural similarity between $u$ and $v$. The graphlet degree vectors are calculated using the orbit counting algorithm [54], are independent of node attributes, and provide a highly constraining measure of local graph topology. We test whether the revised GNNGUARD can defend GNNs trained on graphs with heterophily.

**Experiments.** We synthesize cycle graphs with attached house shapes (see an example in Figure 3), where labels are defined by nodes' structural roles [25]. The synthetic graphs contain 1,000 nodes (no node features, but each node has a 73-dimensional graphlet vector), 3,200 undirected edges, and 6 node labels (*i.e.*, distinct structural roles). We use the strongest performing attacker Nettack-Di to manipulate each graph. Results are shown in Table 5. We find that GNNGUARD achieves the highest accuracy of 77.5%. In contrast, GNN performance without any defense is at most 45.0%. GNNGUARD outperforms the strongest baseline by 19.2%, which is not surprising as existing GNN defenders cannot defend graphs with heterophily. Taken together, these results show the effectiveness of GNNGUARD, when used together with an appropriate similarity function, for graphs with either homophily or heterophily.

## 6 Conclusion

We introduce GNNGUARD, an algorithm for defending graph neural networks (GNN) against poisoning attacks, including direct targeted, influence targeted, and non-targeted attacks. GNNGUARD mitigates adverse effects by modifying neural message passing of the underlying GNN. This is achieved through the estimation of neighbor relevance and the use of graph memory, which are two critical components that are vital for a successful defense. In doing so, GNNGUARD can prune likely fake edges and assign less weight to suspicious edges, a principle grounded in network theory of homophily. Experiments on four datasets and across five GNNs show that GNNGUARD outperforms existing defense algorithms by a large margin. Lastly, we show how GNNGUARD can leverage structural equivalence and be used with heterophily graphs.

## Broader Impact

**Impacts on graph ML research.** Graphs are universal structures of real-world complex systems. Because of strong representation learning capacity, GNNs have brought success in areas, ranging from disease diagnosis [10] and drug discovery [35] to recommendation system [55]. However, recent studies found that many GNNs are highly vulnerable to adversarial attacks [56]. Adversarial attackers inject imperceptible changes into graphs, thereby fooling downstream GNN classifiers into making incorrect predictions [9]. While there is a rich body of literature on adversarial attacks and defense on non-graph data (*e.g.*, text [39], and images [56]), much less is known about graphs. In an effort towards closing this gap, this paper introduces GNNGUARD, a powerful GNN defender that can be straightforwardly integrated into any existing GNN. Because GNNGUARD works with any GNN model, its impact on graph ML research is potentially more substantial than that of introducing another, albeit presumably more robust, GNN model.

**A variety of impactful application areas.** GNNGUARD can be used in a wide range of applications by simply integrating GNNGUARD with a GNN model of user choice that is most suitable in a particular application, as we demonstrate in this paper. Further positive impacts of GNNGUARD include the following. First, we envision that GNNGUARD will help users (*e.g.*, governments, companies, and individuals) avoid potential losses that are caused by misjudgments made by attacked GNNs (*e.g.*, in the face of a massive attack on a financial network) [9]. Second, it would be interesting to explore the possibility of deploying GNNGUARD for key GNN applications in biomedical domain, where, for example, a GNN diagnostics system could predict false diagnosis if it was trained on the attacked knowledge graph [57]. Finally, our model has implications for fairness and explainability of GNNs [36]), which is key to increase users' trust in GNN predictions. Lastly, GNNGUARD can be used for debugging GNN models and understanding of black-box GNN optimization.

**The need for thoughtful use of GNNGUARD.** It is possible to think of a situation where one would use GNNGUARD to get insights into black-box GNN optimization and then use those insights to improve existing attack algorithms, thereby identifying and potentially exploiting new, currently unknown vulnerabilities of GNNs. Because of this possibility and the fact that GNNs are becoming increasingly popular in real-world ML systems, it is important to conduct research to get insights into possible attacks and defense of GNNs.

## Acknowledgments and Disclosure of Funding

This work is supported, in part, by NSF grant nos. IIS-2030459 and IIS-2033384, and by the Harvard Data Science Initiative. The content is solely the responsibility of the authors.

## Footnotes

[1]Code and datasets are available at https://github.com/mims-harvard/GNNGuard.

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
