[Supplementary Material]

# Appendices to "GNNGUARD: Defending Graph Neural Networks against Adversarial Attacks"

**Xiang Zhang**
Harvard University
xiang_zhang@hms.harvard.edu

**Marinka Zitnik**
Harvard University
marinka@hms.harvard.edu

## Appendix A    Defense Performance Against Influence Targeted Attacks

Results are shown in Table 6. We find that the proposed GNNGUARD achieves the best defensive performance against influence targeted attack across five GNN models and four datasets. In particular, GNNGUARD outperforms state-of-the-art defense models by 8.77% on average. Furthermore, compared to the case where the GNN is attacked without any defense, GNNGUARD brings a significant accuracy improvement of 22.6% on average. Remarkably, results show that even most recently published GNNs (*e.g.*, GraphSAINT [23]) are sensitive to adversarial perturbations of the graph structure (cf. "Attack" vs. "No Attack" columns in Table 6), yet GNNGUARD can successfully defend GNNs against influence targeted attacks and can restore their performance to levels comparable to learning on clean, non-attacked graphs.

**Table 6:** Defense performance (multi-class classification accuracy) against influence targeted attacks.

| Model | Dataset | No Attack | Attack | GNN-Jaccard | RobustGCN | GNN-SVD | GNNGUARD |
|-------|---------|-----------|--------|-------------|-----------|---------|----------|
| GCN | Cora | 0.826 | 0.410 | 0.520 | 0.605 | 0.425 | **0.665** |
| | Citeseer | 0.721 | 0.435 | 0.675 | 0.575 | 0.615 | **0.745** |
| | ogbn-arxiv | 0.667 | 0.545 | 0.615 | 0.620 | 0.445 | **0.725** |
| | DP | 0.682 | 0.475 | 0.550 | 0.565 | 0.460 | **0.655** |
| GAT | Cora | 0.827 | 0.425 | 0.550 | 0.605 | 0.450 | **0.635** |
| | Citeseer | 0.718 | 0.510 | 0.675 | 0.575 | 0.615 | **0.815** |
| | ogbn-arxiv | 0.669 | 0.635 | 0.525 | 0.620 | 0.505 | **0.675** |
| | DP | 0.714 | 0.470 | 0.540 | 0.565 | 0.570 | **0.645** |
| GIN | Cora | 0.831 | 0.525 | 0.635 | 0.605 | 0.615 | **0.775** |
| | Citeseer | 0.725 | 0.480 | 0.675 | 0.575 | 0.630 | **0.845** |
| | ogbn-arxiv | 0.661 | 0.570 | 0.605 | 0.620 | 0.525 | **0.710** |
| | DP | 0.719 | 0.505 | 0.585 | 0.565 | 0.605 | **0.695** |
| JK-Net | Cora | 0.834 | 0.525 | 0.665 | 0.605 | 0.625 | **0.755** |
| | Citeseer | 0.724 | 0.485 | 0.675 | 0.575 | 0.610 | **0.865** |
| | ogbn-arxiv | 0.678 | 0.545 | 0.580 | 0.620 | 0.475 | **0.720** |
| | DP | 0.726 | 0.495 | 0.635 | 0.565 | 0.590 | **0.685** |
| Graph SAINT | Cora | 0.821 | 0.405 | 0.495 | 0.610 | 0.395 | **0.645** |
| | Citeseer | 0.716 | 0.460 | 0.665 | 0.590 | 0.605 | **0.735** |
| | ogbn-arxiv | 0.683 | 0.525 | 0.595 | 0.615 | 0.570 | **0.705** |
| | DP | 0.739 | 0.435 | 0.615 | 0.645 | 0.575 | **0.675** |

## Appendix B   Defense Performance Against Non-Targeted Attacks

Results are shown in Table 7. To evaluate how harmful non-targeted attacks can be for GNNs, we first give results without attack and under attack (without defense), *i.e.*, "Attack" vs. "No Attack" columns in Table 7. We also show defense performance of GNNGUARD relative to state-of-the-art GNN defense techniques. First, we find that non-targeted attacks can have a considerable negative impact on the performance of the GNNs. The accuracy of even the strongest GNN is reduced by 18.7% on average. In addition, results show that our GNNGUARD outperforms baselines in most experiments and improves upon baselines considerably. Experiments indicate the proposed GNN defender can successfully mitigate negative impacts brought forward by non-targeted attacks on graphs.

**Table 7:** Defense performance (multi-class classification accuracy) against non-targeted attacks.

| Model | Dataset | No Attack | Attack | GNN-Jaccard | RobustGCN | GNN-SVD | GNNGUARD |
|-------|---------|-----------|--------|-------------|-----------|---------|----------|
| GCN | Cora | 0.826 | 0.578 | 0.684 | 0.571 | 0.678 | **0.714** |
|  | Citeseer | 0.721 | 0.601 | 0.646 | 0.583 | 0.668 | **0.681** |
|  | ogbn-arxiv | 0.667 | 0.410 | 0.409 | 0.436 | 0.413 | **0.444** |
|  | DP | 0.682 | 0.487 | 0.513 | 0.528 | 0.493 | **0.539** |
| GAT | Cora | 0.827 | 0.566 | 0.691 | 0.571 | 0.681 | **0.718** |
|  | Citeseer | 0.718 | 0.676 | 0.667 | 0.583 | 0.680 | **0.699** |
|  | ogbn-arxiv | 0.669 | 0.420 | 0.428 | 0.436 | 0.433 | **0.432** |
|  | DP | 0.714 | 0.519 | 0.548 | 0.528 | 0.534 | **0.566** |
| GIN | Cora | 0.831 | 0.588 | 0.702 | 0.571 | 0.692 | **0.722** |
|  | Citeseer | 0.725 | 0.565 | 0.638 | 0.583 | 0.615 | **0.711** |
|  | ogbn-arxiv | 0.661 | 0.424 | 0.459 | 0.436 | 0.459 | **0.486** |
|  | DP | 0.719 | 0.537 | 0.559 | 0.528 | 0.513 | **0.571** |
| JK-Net | Cora | 0.834 | 0.615 | **0.726** | 0.571 | 0.683 | 0.713 |
|  | Citeseer | 0.724 | 0.574 | 0.647 | 0.583 | 0.679 | **0.698** |
|  | ogbn-arxiv | 0.678 | 0.433 | 0.419 | 0.436 | 0.443 | **0.457** |
|  | DP | 0.726 | 0.486 | 0.537 | 0.528 | 0.541 | **0.587** |
| Graph SAINT | Cora | 0.821 | 0.657 | 0.617 | 0.659 | 0.647 | **0.705** |
|  | Citeseer | 0.716 | 0.628 | 0.596 | 0.637 | 0.652 | **0.659** |
|  | ogbn-arxiv | 0.683 | 0.394 | 0.428 | 0.563 | 0.533 | **0.583** |
|  | DP | 0.739 | 0.473 | 0.572 | 0.499 | 0.524 | **0.537** |

## Appendix C   Classification Accuracy on Clean (*i.e.*, Non-attacked) Datasets with and without GNNGUARD

Next, we want to investigate whether the GNN defender can harm the performance of the underlying GNN if the defender is used on clean, non-attacked graphs. Note that this is a practically important question, as in practice, users might not know a priori whether malicious agents have altered their graph datasets. Because of that, it is essential that a successful GNN defender does not decrease the predictive performance of the GNN in cases when GNNGUARD is turned on, but there is no attack. From results in the main paper and Appendix A-B, we already know that GNNGUARD can defend GNNs when they are attacked. Here, we show that GNNGUARD does not hinder GNNs even when they are not attacked.

Results are shown in Table 8. We observe that GNNs, trained on clean datasets, yield approximately the same performance irrespective of whether a GNN integrates GNNGUARD defense or not. These results suggest that the use of GNNGUARD does not reduce GNN's expressive power or its representation capacity when there are no adversarial attacks.

## Appendix D   Further Details on Datasets

GNNGUARD implementation as well as all datasets and the relevant data loaders are available at https://github.com/mims-harvard/GNNGuard. We provide further dataset statistics in Table 9.

**Table 8:** Classification accuracy on clean (*i.e.*, non-attacked) datasets with and without GNNGUARD.

| | Cora-CLEAN | | Citeseer-CLEAN | | ogbn-arxiv-CLEAN | | DP-CLEAN | |
|---|---|---|---|---|---|---|---|---|
| | w/o | w | w/o | w | w/o | w | w/o | w |
| GCN | 0.826 | 0.817 | 0.721 | 0.716 | 0.667 | 0.683 | 0.682 | 0.681 |
| GAT | 0.827 | 0.829 | 0.718 | 0.719 | 0.669 | 0.674 | 0.714 | 0.717 |
| GIN | 0.831 | 0.832 | 0.725 | 0.726 | 0.661 | 0.671 | 0.719 | 0.716 |
| JK-Net | 0.834 | 0.829 | 0.724 | 0.727 | 0.678 | 0.682 | 0.726 | 0.731 |
| GraphSAINT | 0.821 | 0.819 | 0.716 | 0.721 | 0.683 | 0.669 | 0.739 | 0.727 |

**Table 9:** Dataset statistics. $N$, $E$, $M$, and $C$ denote the number of nodes, edges, node feature dimensionality, and the number of labels/classes, respectively.

| Dataset | N | E | M | C | Node features |
|---|---|---|---|---|---|
| Cora | 2,485 | 5,069 | 1,433 | 7 | Binary |
| Citeseer | 2,110 | 3,668 | 3,703 | 6 | Binary |
| ogbn-arxiv | 31,971 | 71,669 | 128 | 40 | Continuous |
| DP | 22,552 | 342,353 | 73 | 519 | Continuous |
| Synthesized | 1,000 | 3,200 | - | 6 | - |

The new, Disease Pathway (DP) [51] dataset describes a system of interacting human proteins whose malfunction collectively leads to a variety of diseases. Nodes in the network represent human proteins and edges indicate protein-protein interactions. The raw dataset is available at `http://snap.stanford.edu/pathways`. The task is to predict for every protein node what diseases (*i.e.*, labels/classes) that protein might cause. The dataset has 73-dimensional continuous node features representing graphlet-orbit counts (*i.e.*, the number of occurrences of higher-order network motifs), which we normalize via z-scores. This is a multi label node classification dataset. We select 10 most-common labels (diseases), reformulate the task as 10 independent balanced binary classification problems and report average performance across multiple independent runs. The first four datasets are homophily graphs while the last synthesized graph is heterophily graph with structural equivalence. We randomly split the dataset into training (10%), validation (10%), and test set (80%) following the experimental setup in [8].

## Appendix E  Further Details on Hyperparameter Setting

To select hyperparameters and GNN model architectures, we closely follow original authors' guide-lines and relevant papers on GNNs (GCN [3], GAT [21], GIN [7], JK-Net [22], and Graph-SAINT [23]), baseline defense algorithms (GNN-Jaccard [17], RobustGCN [19], and GNN-SVD [18]), and models for generating adversarial attacks (Nettack-Di [8], Nettack-In [8], and Mettack [30]).

We use PyTorch DeepRobust package (`https://github.com/DSE-MSU/DeepRobust`) [58] to implement adversarial attack models and baseline defense algorithms, and PyTorch Geometric package (`https://github.com/rusty1s/pytorch_geometric`) to implement and train GNN models. In all experiments, we set the number of epochs to 200 and use early stopping (we stop training if validation accuracy does not increase for 10 consecutive epochs). We repeat every experiment 5 times and report average performance across independent runs. We set $P_0 = 0.5$, $K = 2$, $D_2 = 16$, and dropout rate as $0.5$, optimize cross-entropy loss using Adam optimizer and learning rate of $0.01$. For other parameters, we follow the setup in [8].