[Reviews · NeurIPS 2020]

Review 1

Summary and Contributions: The authors propose an approach for defending GNNs against poisoning attacks which can be applied to a large family of message passing GNNs. The approach is simple and the authors do a good job of motivating all of the components that they introduce. In short, they assign importance weights to each edge in the graph based on the cosine similarity between the hidden representations of its incident nodes. The main weaknesses of the paper is that the authors do not explore whether the proposed defense is effective against an adaptive attacker. I am inclined to increase my score if they authors show a successful defense with an adaptive attack.

Strengths: * The empirical evaluation is comprehensive though limited in scope (see weaknesses). They compare against three other defense methods on five GNNs, and in these comparisons they outperform their competitors by a good margin. * The approach is general and can be used with different GNNs, and potentially in conjunction with other defenses. * I appreciate the ablation study to justify the addition of the layer-wise memory component.

Weaknesses: * The proposed approach is heuristic. The main downside of heuristic defenses (unlike certified defenses) is that they are often easily broken by an adaptive attacker. With this in mind a proper evaluation of a heuristic defense warrants a strong attempt to break it (by the authors) by proposing an attack that's tailored specifically for it. Without such evidence it is not clear whether this defense would be useful in practice. For example, it is relatively straightforward to add an additional term in Mettack's loss that encourages adversarial edges between nodes with similar representations. * Since their approach relies on the similarity of hidden representations which is correlated with the similarity of the node features, an adaptive attacker can take this into account and modify the features to avoid detection. This limitation should be discussed. * As another adaptive strategy the authors can easily modify Mettack, adding the importance scores and computing the meta-gradients for the modified (rather than the standard) GCN. * The comparison with the baselines is for a single setting of the hyperparameters. Table 3 (right) shows the performance of GNNGuard for different budgets but there is no comparison with other baselines. Since it's not clear what is a reasonable threat model in practice comparison for e.g. different budgets would be insightful. * The defense is limited to networks with homophily. This is however, clearly stated by the authors and a reasonable assumption in practice.

Correctness: The approach is sounds and well motivated.

Clarity: The paper is well written overall and easy to read.

Relation to Prior Work: The authors cover most of the relevant work in the literature and the contributions in relation to previous work are clear.

Reproducibility: Yes

Additional Feedback: * Is learning the memory parameter beta necessary, and if so what is the gain over a pre-specified value? It would be interesting to see what the final learned value was for different datasets. * Is the additional step of edge pruning necessary? * While the approach is well motivated for the poisoning setting it should be applicable without changes to the evasion setting as well. Adding evasion experiments would strengthen the paper. * Experiments with additional attacks (e.g. node addition) and comparison with more baselines should be considered. See https://github.com/safe-graph/graph-adversarial-learning-literature for a detailed overview of attacks and defenses. * Typo on Line 11: how best assign -> how to best assign


Review 2

Summary and Contributions: It proposes to enhance GNNs' robustness against training-time adversarial attacks by introducing a homophily-based attention module (Eq. 3). It further dynamically prunes the edges during training based on the attention scores (Eq. 4-5) and stabilizes training based on memory (Eq. 6) (which actually looks more like momentum update to me.).

Strengths: 1. An important research topic. 2. An easy-to-implement solution. 3. The idea of stabilizing training based on memory is interesting (though it is arguably similar to momentum update).

Weaknesses: W1: Lack of theoretical guarantees. W2: The proposed method makes GNNs only applicable to data and tasks where the dominant factor is homophily. Likely, the proposed method would greatly affect GNNs' performance when dealing with other tasks where complex graph topology features are important. Note that real-world graphs have complex patterns beyond homophily, and sometimes even involve heterophily. W3: The ablation study does not prove that hard pruning of the edges (Eq. 4-5) is necessary. It is possible that once we remove Eq. 4-5 (hard pruning), then Eq. 6 (memory) would become unnecessary as well.

Correctness: Yes.

Clarity: Yes.

Relation to Prior Work: Yes.

Reproducibility: Yes

Additional Feedback:


Review 3

Summary and Contributions: This work presensts a GNN defense approach against edge perturbations of graphs. The approach, termed GNNGuard, learns how best assign higher weights to edges connecting similar nodes while prunning edges between unrelated nodes. Two components are presented: neighbor importance estimation and the layer-wise graph memory.

Strengths: - Empirically experimental results show that GNNGuard outperforms existing defense methods by 15.3% on average.

Weaknesses: - Though components of GNNGuard are mostly motivated by the network homophily theory, but it seems that the design choice seems more like heuristics. It would be great to have some theoretical justifications.

Correctness: - the methodology in this paper is clearly defined. The way to design defense method makes sense to me - I believe in the emperical performance, but it is better to also release the trained models. - A minor typo: in Algorithm 1, “Neural message passsing” h_u^k --> h_v^k

Clarity: The presentation is clear and techniccal details are well organized.

Relation to Prior Work: Related work in adversarial defense for graphs is discussed through line 237-250, line 91-105. It seems that these two parts have certain overlaps and can be merged somehow.

Reproducibility: Yes

Additional Feedback:


Review 4

Summary and Contributions: This paper studies defense strategies against poisoning attacks towards graph neural network models. The key contribution of this paper is a model-agnostic defense algorithm called GNNGuard. GNNGuard has two novel components. The first one uses cosine similarity between nodes to estimate whether an edge between the two nodes is real or fake. Because the first component may vary weights of a layer drastically, this makes the training unstable. Therefore, the second one softens the estimation and pruning strategy of the first component by a weighted combination of the estimated similarity of the current and previous layers.

Strengths: 1. The problem of defense against adversarial attacks towards GNNs is interesting and of practical importance. It has the potential of attracting interest to a large audience. 2. The defense techniques proposed, though simple, are yet novel and effective. The model-agnostic property of GNNGuard is also a desirable feature. 3. The proposed GNNGuard algorithm achieves state-of-the-art performance in defending against poisoning attacks to GNN models.

Weaknesses: 1. My major concern is the assumption that the entire method is built-on, that is, nodes with similar features (embeddings) are more likely to be linked. Whereas this is the case in most graphs and is pointed out in related works, it may not be true in some special graphs. It will be super helpful if the authors can have some discussions on this point. 2. In general, it seems that the pruning strategy of different layers in the GNN is independent. However, intuitively if an edge in one layer is pruned, isn't it supposed to be pruned too on other layers? Are there any such observations made in the empirical evaluations? Or can the authors explain why such correlations are not considered. 3. This is not a major concern to me though -- the paper does not compare with [7]. I understand that [7] uses clean graphs as training data. However, the authors should justify why [7] is not compared, or compare with it anyway. Even if [7] sometimes outperforms GNNGuard I will not be surprised and this does not deteriorate the value of GNNGuard. Spotted a typo at the beginning of Section 3: “Let … denotes” -> denote

Correctness: The claims, methods are correct to me subject to that the assumption (refer to weaknesses point 1) made is validated.

Clarity: The paper is well written, with a nice discussion of related work.

Relation to Prior Work: Yes.

Reproducibility: Yes

Additional Feedback: Thanks for the rebuttal from the authors. The rebuttal clarifies the weaknesses points 2 & 3. However, I am not entirely convinced by point 1. A major concern as also pointed out by Reviewers #2 #4, is that the proposed method works only on graphs with homophily, which essentially assumes the node similarity implies edges between nodes. Although the rebuttal gives an example where structural similarity of nodes can also be used to design defense strategies, it is still some sort of assumption on the graph. Such assumption limits the generalization of GNNGuard. Therefore I am not altogether convinced for that point will keep my score unchanged. PS: It would be super helpful if the authors can open source their code for implementation.

[Author Response · NeurIPS 2020]

We thank the reviewers for their time and valuable feedback. Overall, we are glad that the reviewers found our
GNNGUARD to be *"novel and effective"*, *"model-agnostic"*, *"of practical importance"*, *"does a good job of motivating*
*all of the components"*, has an interesting *"idea of stabilizing training"*, and *"achieves state-of-the-art performance."*
Below, we clarify several important points raised by the reviewers. These issues are mainly caused by the omission of
certain details due to the limited space. An extra page in the final version will allow us to include the requested details.
We believe these clarifications, together with new analyses, resolve all key issues raised.

**(1) GNNGUARD can defend graphs with complex patterns beyond homophily. R2, R4**, and

**GNNGuard can work on graphs with heterophily**

Structural roles

**R5** raise a critical point that GNNGUARD is limited to graphs with homophily. While homophily
is a very common assumption in existing research, as nicely pointed out by **R1**, *"clearly stated by*
*the authors and a reasonable assumption in practice"*. We would like to clarify that GNNGUARD
can defend against attacks on graphs with heterophily. In fact, **it is straightforward to use GNN-**
**GUARD on graphs with heterophily where connected nodes do not necessarily share similar**
**labels/attributes but share similar roles/positions in the graph (see Figure)**. In response to
reviewers, we show how to use GNNGUARD on graphs with structural roles, a prominent type
of heterophily. To do this, we replace the cosine similarity (L183, P4) with a graphlet degree similarity [Milenković
et al, Cancer Inform.'08], which quantifies structural similarity between nodes in terms of their structural properties,
e.g., triangles, betweenness, stars, etc. The graphlet degree similarity is independent of node attributes [Sarajlic et al,
Sci. Rep'16] and provides a highly constraining measure of local topology. In the experiment, we synthesized cycle
graphs ($N = 1000$, $E = 1600$, $C = 6$) with attached house shapes (see a toy example in Figure), in which node labels
are defined by nodes' structural roles [Donnat et al, KDD'18]. We then run GNNGUARD with the following setup:
underlying GNN: GIN; attacker: Nettack-Di. We find that GNNGUARD achieves accuracy of 77.5%. In contrast, GNN
performance without any defense is only 45%. Further, GNNGUARD outperforms the strongest baseline by 19.2%,
which is not surprising as existing GNN defenders cannot defend graphs with heterophily. These new results indicate
that GNNGUARD, used in a combination with an appropriate similarity function, can work on graphs with heterophily.
**(2) Edge pruning. (2.1) R1** and **R2** raise a concern that our ablation does not examine whether edge pruning
(Eq. 4-5) is necessary or not. We conduct new experiments showing that edge pruning is a necessary compo-
nent of GNNGUARD. By removing edge pruning, GNNGUARD's performance drops by 8% on average (see
Table) using the setup: dataset: orgb-arxiv; underlying GNN: GIN; attacker: Nettack-Di. Further, edge pruning
get even more important when we use GNNGUARD on graphs with heterophily because pruning of adversarial
edges has a direct effect on the choice of structural similarity between nodes (e.g., graphlet degree similarity).
**(2.2) R5** requests a clarification of edge pruning across GNN

| Model | No Defense | **w/o pruning** | w/o memory | GNNGUARD |
|---|---|---|---|---|
| GCN | 0.235 | 0.350 | 0.405 | 0.425 |
| GAT | 0.210 | 0.315 | 0.475 | 0.520 |
| GIN | 0.315 | 0.540 | 0.610 | 0.640 |
| JK-Net | 0.335 | 0.565 | 0.625 | 0.635 |
| GraphSAINT | 0.245 | 0.305 | 0.360 | 0.375 |

layers. We note that if an edge is pruned in one layer, it will
get pruned in all subsequent layers. This is because we quantify
similarity $s_{uv}^k$ between $u$ and its neighbor $v$ in each layer (L182-
183, P4). Suppose edge $e_{uv}$ is pruned in the (k-1)-th layer, then $v$
is no longer a neighbor of $u$ in the subsequent layers. This means that edge pruning is multi-layer interdependent, and
thus GNNGUARD has strong control over the exchange of neural messages in a GNN.
**(3) GNNGUARD can defend against adaptive attacks. R1** raises an important point regarding the threat of adaptive
attacks. We conduct new experiments to evaluate GNNGUARD's ability to defend against adaptive attacks. We develop
an Adaptive-Mettack which encourages adversarial edges between nodes with similar representations. In specific, we
add the cosine similarity of node-pair $(u, v)$ when calculating the score function $S(u, v)$ [25]. The attacker will select
the edge with the highest score as the adversarial edge. On Cora [setup: Cora, GIN, Mettack with 20% budget], we
find that the accuracy of GNNGUARD (0.714) surpasses all the baselines including w/o-defense (0.653), GNN-Jaccard
(0.679), RobustGCN (0.571), and GNN-SVD (0.683). On Citeseer, GNNGUARD (0.658) also outperforms w/o-defense
(0.574), GNN-Jaccard (0.598), RobustGCN (0.583), and GNN-SVD (0.607). The new results show that GNNGUARD
can defend GNN models against adaptive attacks. We will carefully discuss the analysis in the final version.
**(4) Attacks with varying budgets. R1** raises an important point on examining defense performance with varying
budgets. We conducted experiments and can share one of more insightful observations. Non-targeted attacks (e.g.,
Mettack) are more sensitive to budget amounts than targeted attacks (e.g., Nettack). Mettack causes slight harm within
2% of the budget, but it becomes more harmful when the budget exceeds 10% of the graph size (i.e., number of edges).
We find that GNNGUARD consistently outperforms all baselines. We will provide full results in the final version.
**(5) Baselines and further clarifications. (5.1)** We thank **R1** for sharing an interesting survey [Sun et al, arXiv'18],
which we have studied extensively and will include it in our final version. We will also carefully discuss other types of
attacks (*e.g.*, injecting vicious nodes) and defenders (*e.g.*, defending via VAE). **(5.2) R5** asks us to justify why ref. [7] is
not compared to GNNGUARD. This is a misunderstanding as ref. [7] introduces a GNN attacker, not a GNN defender.
Our GNNGUARD is a GNN defender, and thus the comparison is not possible. However, we did use the attacker from
[7] (L259-260, P7) to extensively evaluate how successful GNNGUARD's defense is. GNNGUARD achieves strong
performance (see Table 1 and Appendix A). We will provide detailed clarifications in our final version.

[Meta-Review · NeurIPS 2020]

Three reviewers participated in the discussion. The main concern was that the proposed method works only on graphs with homophily. Although the rebuttal gives an example where structural similarity of nodes can also be used to design defense strategies, it is still some sort of assumption on the graph. That said, some reviewers pointed out that this is not a deal-breaking limitation, since the assumptions are clearly stated in the paper. The reviewers also agreed that the proposed method is simple yet effective. However the lack of theoretical guarantees is another limitation that may be interesting to explore in future work.